# Cyclic Voltammetry of Screen-Printed Carbon Electrode Coated with Ag-ZnO Nanoparticles in Chitosan Matrix

**DOI:** 10.3390/ma16083266

**Published:** 2023-04-21

**Authors:** Elena Emanuela Herbei, Petrică Alexandru, Mariana Busila

**Affiliations:** Interdisciplinary Research Centre in the Field of Eco-Nano Technology and Advance Materials CC-ITI, Faculty of Engineering, “Dunărea de Jos” University of Galati, 47 Domnească, 800008 Galati, Romania; elena.herbei@ugal.ro

**Keywords:** surface modified screen-printed electrode, sensors, silver nanoparticles, chitosan matrix

## Abstract

In this paper, the authors describe the fabrication of nanocomposite chitosan-based systems of zinc oxide (ZnO), silver (Ag) and Ag-ZnO. Recently, the development of coated screen-printed electrodes using metal and metal oxide nanoparticles (NPs) for the specific detection and monitoring of different cancer tumors has been obtaining important results. Ag, ZnO NPs and Ag-ZnO prepared by the hydrolysis of zinc acetate blended with a chitosan (CS) matrix were used for the surface modification of screen-printed carbon electrodes (SPCEs) in order to analyze the electrochemical behavior of the typical redox system of a 10 mM potassium ferrocyanide—0.1 M buffer solution (BS). The solutions of CS, ZnO/CS, Ag/CS and Ag-ZnO/CS were prepared in order to modify the carbon electrode surface, and were measured at different scan rates from 0.02 V/s to 0.7 V/s by cyclic voltammetry. The cyclic voltammetry (CV) was performed on a house-built potentiostat (HBP). The cyclic voltammetry of the measured electrodes showed the influence of varying the scan rate. The variation of the scan rate has an influence on the intensity of the anodic and cathodic peak. Both values of currents (anodic and cathodic currents) have higher values for 0.1 V/s (Ia = 22 μA and Ic = −25 μA) compared to the values for 0.06 V/s (Ia = 10 μA and Ic = −14 μA). The CS, ZnO/CS, Ag/CS and Ag-ZnO/CS solutions were characterized using a field emission scanning electron microscopy (FE-SEM) with EDX elemental analysis. The modified coated surfaces of screen-printed electrodes were analyzed using optical microscopy (OM). The present coated carbon electrodes showed a different waveform compared to the voltage applied to the working electrode, depending on the scan rate and chemical composition of the modified electrodes.

## 1. Introduction

There is a big necessity for the detection of cancer tumors molecular markers, whose early recognition is of fundamental importance in successfully treating the disease using miniaturized devices. Electrochemical methods using different shaped nanoparticles as surface modifiers in an SPCE are a low-cost, highly sensitive, fast and time-saving way to determine and screen the presence of different cancer tumors or different compounds with redox activity. Sensors and biosensors have been investigated in the last few decades due to the intensive electrochemical redox activity of proteins and enzymes [1]. Due to the use of different shaped, doped, and undoped nanomaterials as working electrodes, the sensors have high selectivity and sensitivity. Synthesis conditions significantly determine the changes in the morphology and structure of NPs, altering the physical and chemical properties. Due to the growth direction of versatile ZnO ([0001],
[$2\overset{-}{110}$]
and [$01\overset{-}{10}$]), changing synthesis conditions determine a variety of ZnOs with properties that can lead to the use of ZnO particles in different areas, such as electronic device [2].

Gasparotto and coworkers reported a detailed review regarding the utilization of ZnO nanorods-Au NP nanohybrids for the electrochemical immunosensor developed onto a glass substrate [3]. They used sputtered gold (Au) as working and counter electrodes, and silver (Ag) as reference electrodes, electrically isolated from the others. They obtained electrochemical responses with well-defined CV curves for the bioelectrode prepared at all measured concentrations of Au. Pannada et al. used Au-Cu/ZnO nanoflowers for dopamine identification in urine samples [4]. The graphite carbon electrode modified with ZnO nanoflowers exhibited an apparently strong reversible peak which can be assumed to be the anodic and cathodic peaks, and high current peaks. Researchers successfully used another system (immunosensor) with Au/chitosan modified SPCEs for the detection of antigen 125; the target antigen (Ag) was captured by its specific antibody (Ab1), and could then be detected by a marked secondary antibody (Ab2) [5]. 

Faruk et al. used benzothiophene derivative layers on carbon electrodes for monitoring ovarian cancer tumor protein CA 125 [6]. Modified glassy carbon electrodes with zinc oxide, graphene oxide and silver in chitosan matrices were used to monitor the enzymatic behavior of glucose [7]. Additionally, for liver cancer, Ching and coworkers used commercial ZnO (20 nm) nanoparticles to fabricate immunosensors for liver cancer [8]. Magnetite and graphene were used by Feng and coworkers in a modified glassy carbon electrode for the rapid detection of *Salmonella* in milk. Due to the conductivity and mechanical stability of graphene and the large specific surface area of Fe_3_O_4_, the nanocomposite provides better electrical conductivity. In this case, the cyclic voltammetry response current of the electrode modified by the Fe_3_O_4_–graphene significantly increased compared to the unmodified GCE. The gold nanoparticles added in the magnetite–graphene mixture amplified the current signals [9]. Carbon doped with nitrogen for the modification of a glassy carbon electrode (GCS) was used by Karikalan for the identification of caffeic acid in red wine, reaching a detection limit of 0.0024 μM [10]. To identify one of the common environmental pollutants in the human body, arsenic (known as human carcinogens), metal nanoparticles such as gold, silver and platinum electrodes were used in order to determine the presence of As (III) [11]. Additionally, zirconia nanoparticles in a graphene matrix were used by Lin to detect carcinoembryonic antigens through the modification of a GCE [12]. A complex review presented by Sharifianjazi and coworkers highlighted that carbon nanotubes, graphene, metal nanoparticles, inorganic and magnetic nanoparticle and metal–organic framework-based sensors were applied for the identification of lung and bronchus cancer, breast cancer, prostate cancer and colon cancer [13]. Gazze proved in an original article that using graphene biosensors, ovarian cancers in stage 1 or 2 can be detected using CA125 antigens in 75% of studied cases [14]. Catalytic metals and metal oxides at nanoscale with different shapes (nano-urchins, nanowires, nanoflowers and nanoplates), such as cobalt, nickel, cooper, iron and noble metals, are used in GCEs as sensors and biosensors for the identification of glucose, hydrogen oxidase, dopamine, catechol, oxalic acid, L-dopa, 4-aminophenol, hydrazine and ascorbic acid [15].

Therefore, in general, metal and metal oxide at nanoscale are used in GCEs and SPCEs to modify the active coated surface of the working electrode to obtain electrochemical signals for different chemical compounds from human bodies to identify disease signals.

Surface modifications of SPCEs with biopolymers as chitosan are used nowadays. The use of chitosan is very important due to its biocompatible properties, which allows them to be used in direct contact with the human body [16]. Nainggolan used chitosan in combination with zinc oxide to measure the cyclic voltammetry on a cooper electrode. The cyclic voltammetry showed oxidation and the reduction reaction was very stable in the presence of the cooper electrode. The combination of ZnO nanoparticles and chitosan showed a good sensitivity, which leads us to believe that ZnO nanoparticles can be used as sensitive compounds for medical research to identify the presence of different markers [17]. Almeida used chitosan with gold nanoparticles in order to determine bisphenol A compounds, which are considered to be endocrine-disrupting compounds with toxicological effects, even at low doses [18]. The electrochemical measurement of fluorine-doped tin oxide electrodes showed the oxidation and reduction process. Another study by Martínez-Huitle that used different concentrations of chitosan solution for the surface modification of GCEs proved that by changing the chitosan solution used in the preparation procedure, a charge-based molecular recognition of ionic species is possible [19]. A composite of graphene oxide/cobalt/chitosan was used in SPCEs for electrochemical measurements in order to determine the various concentration of D-glucose and showed a sensitivity of 15 mM [20]. The same system, but with cooper nanoparticles (Cu), was studied by Renjini to determine the dopamine level an important neurotransmitter [21]. Daclatasvir, a hepatitis C antiviral drug, was identified by a carbon paste electrode modified with chitosan and MWCNT [22].

Regarding the uses of screen-printed carbon electrodes produced by different suppliers, Dascalescu used carbon, mesoporous carbon and ordered mesoporous carbon-based electrodes in order to determine the L Dopa compound from *Mucuna pruriens* seeds using cyclic voltametric determination [23]. In another complex review regarding the serotonin detection from clinical and pharmaceuticals samples, the carbon electrodes were modified with carbon nanotubes, benzofuran, polyalizarin red-S, Nafion/Ni(OH)_2_, graphene oxide and SPE modified with silver and silver selenite nanoparticles [24]. In a review paper, Balaji and Zhang presented the modification of a carbon electrode by decorating with graphene oxide and combining it with a single-strand DNA sequence [25]. The modification with graphene oxide helps in the detection of the cancer breast gene (BRCA 1), and the response of the sensor was given by the increase of the oxidation peaks in the CV determination. Another substantial review by Gajdosova showed that SPCEs modified with reduced graphene oxide (RGO)/MWCNT, graphene oxide, ZrO_2_-RGO, Au Nps-SWCNTs, Au nanoflowers and TiO_2_-nanotubes were used to identify different markers for cancer disease as BRCA1, MUC, recombinant human CA125/MUC16 protein, miRNA-21 and MUC1 and MCF-7 MDA-MB-2 [26].

The cyclic voltammetry technique, which is very popular and simple, is used most often to elucidate electrode mechanisms and to observe the oxidation and reduction peaks. It is an important method because it can detect the presence of cancers, tumors, neurotransmitters and compounds with electrochemical activity in the presence of nanomaterials. Despite being a well-known method, the experimental condition is still in progress for a lot of compounds.

Due to the necessity of the detection of cancer tumors, the early recognition of which is fundamental for treating the disease using miniaturized devices, we started a study of SPCEs that had surfaces modified with zinc oxide nanoparticles, Ag dopped zinc oxide and chitosan. This study aimed to obtain electrical signals from voltammetry measurements to be used in future research on electrical responses for CA-125 antigens identification and recombinant human CA125/MUC16 protein markers. In this paper, we present the uses of CS, ZnO/CS, Ag/CS and Ag-ZnO/CS to modify the surface electrode of SPCE to obtain an electrochemical response regarding the oxidation and reduction response. The novelty of our work in relation to specialized literature consists of the utilization of freshly prepared zinc oxide nanoparticles obtained by hydrolytic method, Ag-ZnO NPs and non-commercial ones to modify the carbon electrode surface with these systems. Here we introduce the house-built potentiostat (HBP) to measure the CV curves of the coated screen-printed electrodes and the experimental steps in which we modified the working electrode with CS, ZnO/CS, Ag/CS and Ag-ZnO/CS solutions. The obtained system was utilized to analyze the electrochemical behavior in a buffer solution (pH = 7) and a potassium ferrocyanide-buffer solution. By covering the graphite electrode with different solutions, we obtained electrochemical signals for each of the four systems studied. The electrochemical responses of SPCEs showed different behavior depending on the electron transfer from electrode to the solution. 

## 2. Materials and Preparation Methods

### 2.1. Reagents and Solutions

The buffer solution (pH-7.010) HI6007 was purchased from Hanna Instruments. Chitosan with a low-molecular-weight (50 kDa) and potassium ferrocyanide were purchased from Sigma-Aldrich Chemical Co. (Bucharest, Romania) As a prepared buffer solution, (PBS) pH = 7 and 10 mM potassium ferrocyanide were used to analyze the electrochemical behavior of the modified sensor behavior. The ZnO/CS, Ag/CS and Ag-ZnO/CS nanocomposites were synthetized according to the protocols established in a previous study by Busila, in which she prepared a 0.035 M solution of zincacetate dyhidrate with adequate amounts of nitrite silver [27]. To have the nanocomposite ready for the first step, the chitosan solution was prepared using 1 g of chitosan in an acid 1% acetic solution. For the preparation of the final solutions used in the surface modification of the carbon working electrodes, the solid nanocomposites (1 mg of each) were dispersed in an ethanol solvent (2 mL), and each sample was maintained under sonication for 30 min at room temperature. Regarding the preparation of the electrochemical solution, 2 mL of PBS was mixed with 2 mL of K_4_[Fe(CN)_6_], in order to have the PBS/K_4_[Fe(CN)_6_] solution for the CV measurements. The solution was used for each modified electrode at different scan rates to see the cyclic voltammetry behavior. The as-synthetized nanocomposites deposited by spin-coating (2000 rpm for 20 s) onto the glass substrates were covered with a gold layer for the SEM analysis. The morphology of the investigated samples was highlighted by scanning electron microscopy (SEM) using a Quanta 200 (FEI) system.

### 2.2. House-Built Potentiostat and Electrochemical Cell

The house-built proposed potentiostat (Figure 1) was practically designed to be a system for future research on the cyclic voltammetry measurements of an immunosensor for the identification of CA-125 antigens and recombinant human CA125/MUC16 proteins. The design scope of was to measure lower currents to identify electrochemical signals. Our device was developed on an open-source electronic board with a simple integrated development for programming the microcontroller. The simple Arduino-based potentiostat was developed in the same format as Crespo and coworkers, and the electrical response was evaluated using different potassium ferricyanide solutions at different scan rates [28]. Different potentiostat structures are presented in the literature; however, our potentiostat system can reach voltage waveforms of approximately 100 Hz, with an amplitude of −0.6 V to 1.5 V and maximum scan rates of 0.7 V/s [29]. In respect to the electrochemical cell, this was made of insulator polymer-poly methyl-methacrylate. As we used a very low volume, the solution was inserted with an o-rubber ring between the slices to prevent solution leakage. The SPCE adaptor used was purchased from Aliexpress on 1 February 2022 (https://www.aliexpress.com/p/order/index.html). Figure 1 presents the system for the CV measurements of SPCEs: the electrode adapter and electrochemical cell (Figure 1a); the potentiostat, and the circuits between the cell and the potentiostant (Figure 1b); and the check chart (Figure 1c) to ensure that the connections between the electrodes work.

### 2.3. Pretreatment Condition

To ensure that the CV of the surface modified-SPCE was stable, the electrodes were activated, and the phase shift would not modify the form of the anodic and cathodic peaks, each measurement was conducted 5 times with different scan rates, from 0.02 to 0.7 V/s. After these 5 cycles, we measured again for 1 additional cycle. In our article, we present the results of the CV with 0.05 V/s for the unchanged SPCE in the PBS, and with 0.1 V/s for the modified electrode in the PBS/K_4_[Fe(CN)_6_] solution.

### 2.4. Custom Made Working Electrode

For the cyclic voltammetry measurements, ceramic substrate screen printed carbon electrodes (SPCEs) were used from the Methrom Drop Sens model C110: L33 × W10 × H0.5 mm [30]. The modified working electrode (WE) was made of carbon (4mm diameter), the auxiliary electrode (AE) was made of carbon, and the reference electrode (RE) was made of silver. For the contact with the potentiostat, a commercial adaptor was used. A further 2 µL of nanocomposite suspension was drop casted on the working electrode (Figure 2) surface of SPCE and dried in warm air from a blower. The excess liquid was absorbed with filter paper, and did not affect the AE and RE.

## 3. Results

### 3.1. Electrochemical Characterization–First Steps in Buffer Solution

The potential graph showing the electrochemical stability of electrolytes is important in order to determine the oxidation and reduction potential.

The electrochemical behavior was studied to determine the influence of surface modified SPCEs in a typical redox analyte. In the first step, we measured the unchanged surface SPCE in a buffer solution to optimize the parameters. To do this, we measured the electrochemical signal of the SPCE at different scan rates (not presented in article) to stabilize the electrode, and then a single cycle at a scan rate (Sr) of 0.05 V/s (Figure 3) in the PBS. 

The voltammograms recorded of the SPCE for the five cycles and one cycle do not preset anodic or cathodic peaks; this, in fact, ensures that the buffer and electrode are not impure [23]. After this measurement, we conducted the CVs for the unchanged SPCE in the 10 mM potassium ferrocyanide-buffer solution. The redox system was used in our further measurements of the surface modified SPCE. For this system, we measured the electrode behavior at 0.06 V/s and 0.1 V/s (Figure 4).

The scan rates of the measured electrode were observed as a pair of well-defined peaks with the same values of anodic and cathodic currents. The redox reaction of the potassium ferrocyanide solution presented an anodic peak because of ferrocyanide oxidation, and the cathodic peak appeared because of the ferricyanide reduction on the WE. The graph shows us that the scan rate had an influence on the intensity of the anodic and cathodic peaks. Both intensities had higher values for 0.1 V/s (Ia = 22 μA and Ic = −25 μA) compared to the values for 0.06V/s (Ia = 10 μA and Ic = −14 μA).

### 3.2. Electrochemical Characterization—PBS/K_4_[Fe(CN)]_6_ Solution

Further CV measurements were conducted on the surface modified SPCEs. For each electrode modified, we measured at a scan rate from 0.02 to 0.7 V/s. In this paper, and for future studied we present, the CVs were only recorded at 0.1 V/s for each surface modified electrode. Figure 5a–d shows the CVs for the CS, ZnO/CS, Ag/CS and Ag-ZnO/CS systems.

In case of the CS modified SPCE (Figure 5a), we observed an increase of the anodic and cathodic intensity peaks due to the increase of electron transfer from the oxidized form to the reduced form of redox analyte (Ia = 45 μA and Ic = −34 μA). For the ZnO/CS modified SPCE, the CV is presented in Figure 5b.

Analyzing Figure 5b with the ZnO/Cs modified electrode, we observed the movement of the anodic and cathodic intensity peaks in relation to the XoY system (Ia = 15 μA and Ic = −13 μA). Going further on the other two electrodes modified with Ag/Cs and Ag-ZnO/Cs, we observed the almost complete disappearance of the anodic peak for Ag/Cs (Figure 5c); in the case of Ag-ZnO/Cs (Figure 5d), both the anodic and the cathodic peaks were hardly noticeable, and this can be assumed to be due to the low rate of electron transfer.

In order to observe the electron transfer rate, we overlapped the CV graph for each of the modified electrodes (Figure 6). We can observe that the electron transfer rate was faster in the case of the Cs modified electrode. The ferricyanide diffusion from the solution to the electrode surface was due to the concentration of determine ferricyanide reduction to ferrocyanide. In the case of the Cs modified SPCE, the voltammogram presented the anodic and cathodic peaks, and in the case of Ag/Cs, the oxidation and reduction were missing. In the case of ZnO/Cs, we had very well-defined oxidation and reduction currents. This can be assumed to be due to the electron transfer rate of the ZnO nanoparticles.

For the HBP, the scan rate was calculated using Equation (1), where *V_max_* and *V_min_* were, respectively, the maximum and minim of voltage; *nc* was number of cycles; *Ff* was a fill factor that varies depending on the signal that the potentiostat shows; and *ts* was the time step which is varied to obtain different scan rate.
(1)$$\mathrm{Scan}\;\mathrm{rate}\;(\mathrm{Sr})=\frac{\left(V_{max}-V_{min}\right)\cdot 1000}{nc\cdot Ff\cdot ts}\;(V/s)\;$$

Depending on the surface modifier, the voltammograms were clear and showed defined oxidation and reduction peaks for CS and ZnO/CS. In the case of Ag/CS, the oxidation peak was very small, almost undetectable, while the reduction peak moved to the right, and decreased to −10 µA. For Ag-ZnO/CS, the oxidation and reduction peak loss was almost unnoticeable. Analyzing the four-system used for the surface modification of each electrode, we conclude that the best behavior can be assumed for the electrodes modified with chitosan and ZnO/Cs 

### 3.3. Morphology Characterization 

Optical images (Figure 7) were analyzed to observe the SPCE surface modified with CS, ZnO/CS, Ag/CS and Ag-ZnO/CS. For CS and Ag/CS, the surface of the layer after solvent evaporation was smooth. The surfaces of the composites blended with ZnO and Ag-ZnO became uneven, with dense agglomeration.

The SEM images with EDX (Figure 8) of the CS (Figure 8a), ZnO/CS (Figure 8b), Ag/CS (Figure 8c) and Ag-ZnO/CS (Figure 8d) of the thin films of solution used for electrode modification, obtained by spin-coating, show a smooth surface in the case of chitosan (Figure 8a). For the nanocomposites system, the surface became ununiform, with agglomeration nanoparticles that ranged between 40 to 100 nm for Ag/CS, ZnO/Cs and Ag-ZnO/Cs (Figure 8a–c). A better dispersion of nanoparticles was observed for the system with silver nanoparticles. 

## 4. Conclusions and Future Perspectives

In this study, a home built potentiostat for cyclic voltammetry measurements, to be used as a future embedded system for immunosensors, was developed. A simple and sensitive method for potassium ferrocyanide identification was tested.

The SEM/EDX images showed the obtained ZnO and Ag nanoparticles dimensions ranging from 40 to 100 nm.

The CVs of the surface modified SPCEs of CS, ZnO/CS, Ag/CS and Ag-ZnO/CS systems in potassium ferrocyanide-buffer solution (pH = 7) were studied. The electrochemical parameters results showed that best behavior is assumed to chitosan surface modified SPCEs and a good behavior for ZnO/CS. 

The development of a fast and low-cost method allows for the frequent screening of patients, helping in an early diagnostic of different types of cancer markers. Therefore, the synthesis of new hybrid/composites nanomaterials with suitable functions should be developed to obtain a better electrochemical response and a targeted answer for specific cancer marker. 

In the oncological field, biosensors based on screen-printed electrodes contribute to the detection of specific biomarkers and that differentiate between normal cells and potentially cancerous cells.

## 5. Patents

The modification of the surface of screen-printed electrodes with silver and gold nanoparticles in chitosan matrices is part of a patent application. Modified immunosensor with hybrid nanostructured material and fast cancer method detection sent in our state office for Invention and Trademarks. 

## Figures and Tables

**Figure 1 materials-16-03266-f001:**
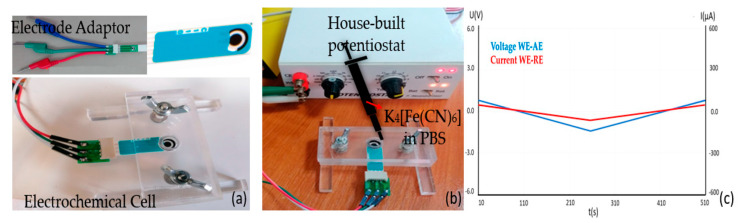
System for cyclic voltammetry measurements: the electrochemical cell (**a**); the potentiostat and the circuits between cell and potentiostat (**b**); check chart (**c**).

**Figure 2 materials-16-03266-f002:**
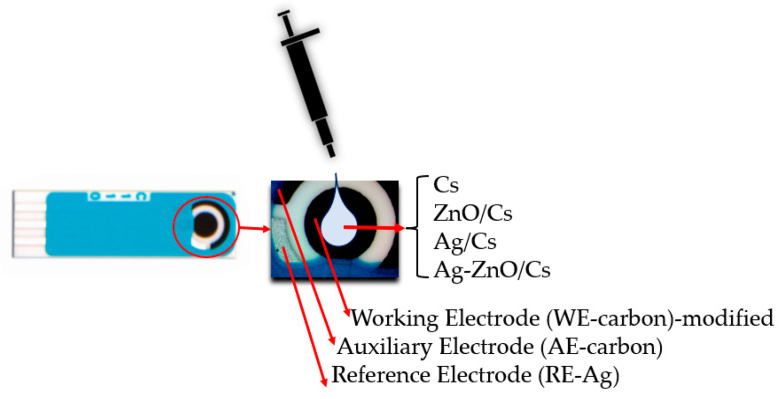
Set-up of modified carbon working electrode.

**Figure 3 materials-16-03266-f003:**
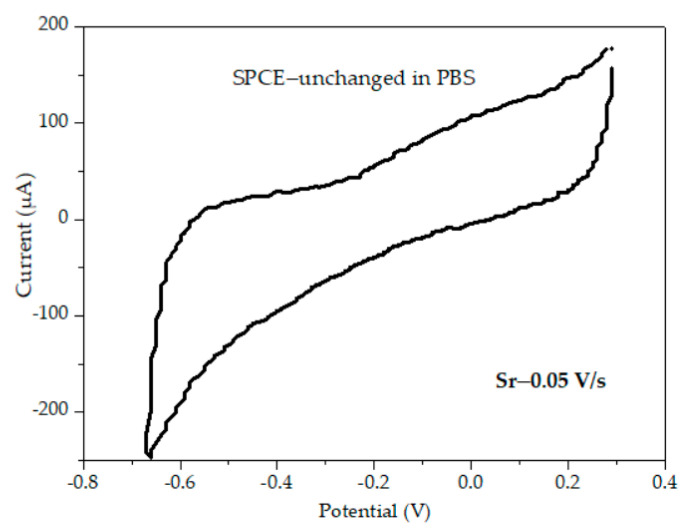
CVs of unchanged SPCE—1 cycle.

**Figure 4 materials-16-03266-f004:**
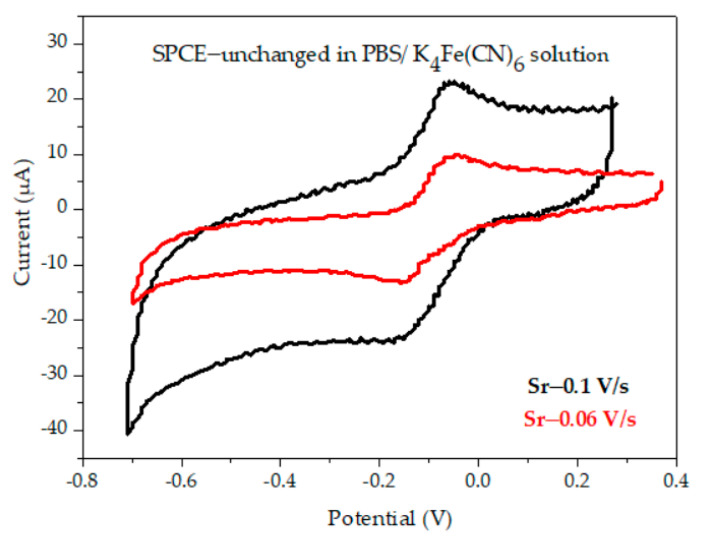
CVs of unchanged SPCE in PBS/ 10 mM K_4_[Fe(CN)]_6_ at 0.06 and 0.1 V/s, with linear dependence of anodic current and square root of Sr.

**Figure 5 materials-16-03266-f005:**
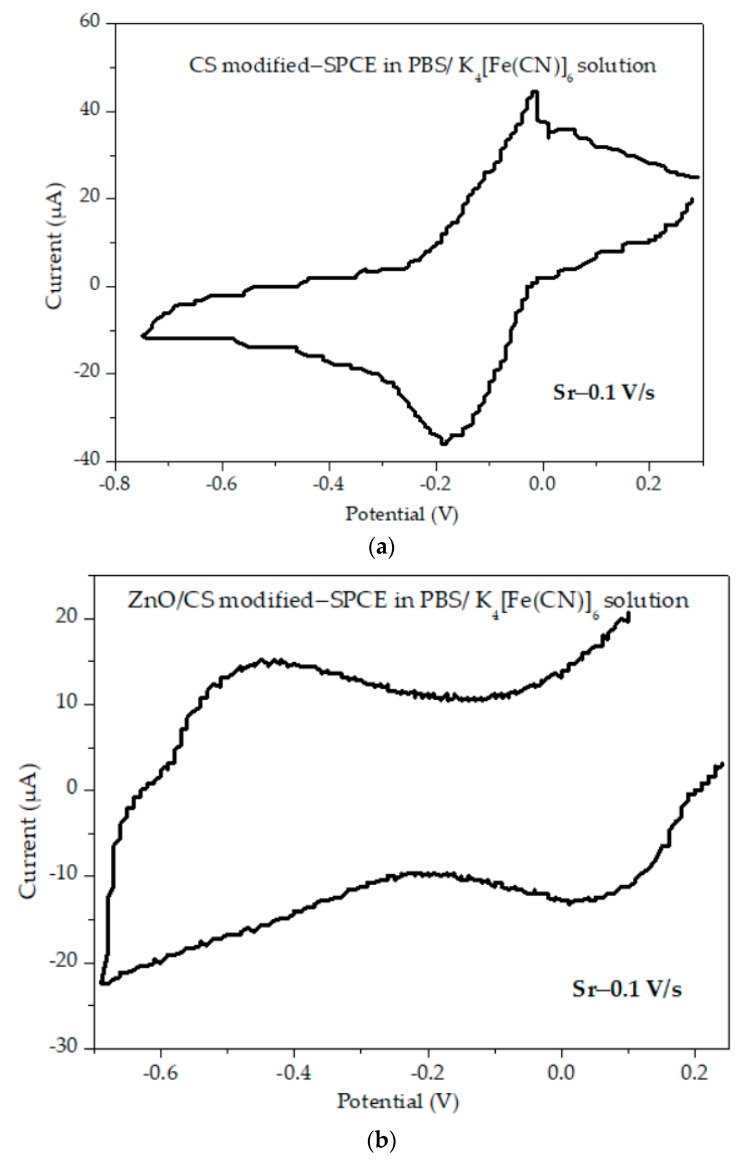
(**a**) CV of CS modified-SPCE in PBS/ 10 mM K_4_[Fe(CN)]_6._; (**b**) CV of ZnO/Cs modified-SPCE in PBS/ 10 mM K_4_[Fe(CN)]_6_.; (**c**) CV of Ag/CS modified-SPCE in PBS/ 10 mM K_4_[Fe(CN)]_6_.; (**d**) CV of Ag-ZnO/CS modified-SPCE in PBS/10 mM K_4_[Fe(CN)]_6_. In the case of Ag/Cs and Ag-ZnO/Cs the low current signals for anodic and cathodic current are highlighted using blue circles. For Ag/Cs modified SPCE only cathodic peak is observed and in the case of Ag-ZnO/Cs the intensity of anodic and cathodic peaks are very low in intensity. Regarding the presence of silver nanoaparticles in the case of these two electrodes we can suppose that the presence of silver blocks the difussion of [Fe(CN)]_6_ and increase the resistance of electrode interface.

**Figure 6 materials-16-03266-f006:**
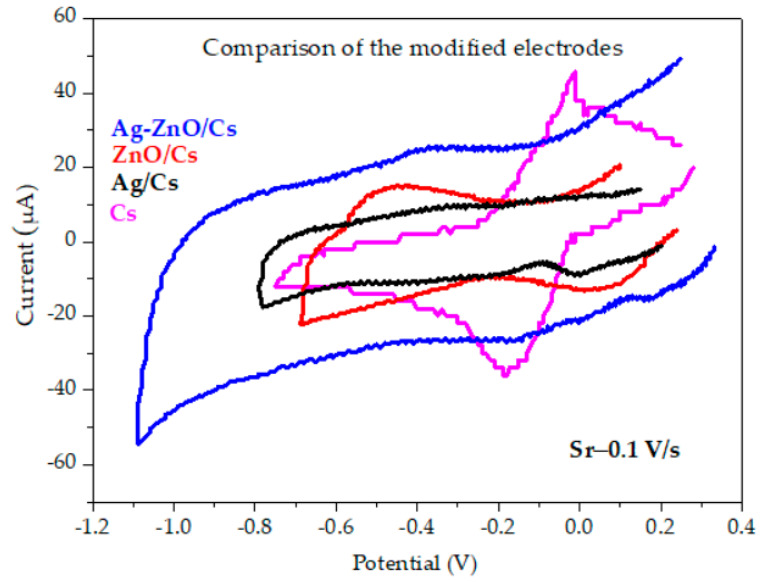
Cyclic voltammograms for the reduction of 10 mM ferricyanide for different modified electrodes at a scan rate of 0.1 V s-for Ag-ZnO/Cs, ZnO/Cs, Ag/Cs and Cs.

**Figure 7 materials-16-03266-f007:**
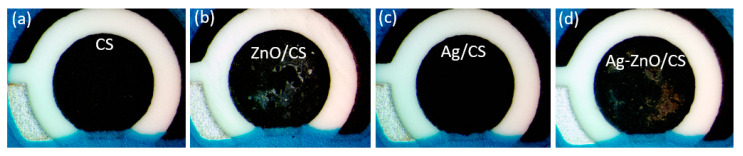
Optical images of surface modified SPCEs with CS (**a**); ZnO/CS (**b**); Ag/CS (**c**); Ag-ZnO/CS (**d**).

**Figure 8 materials-16-03266-f008:**
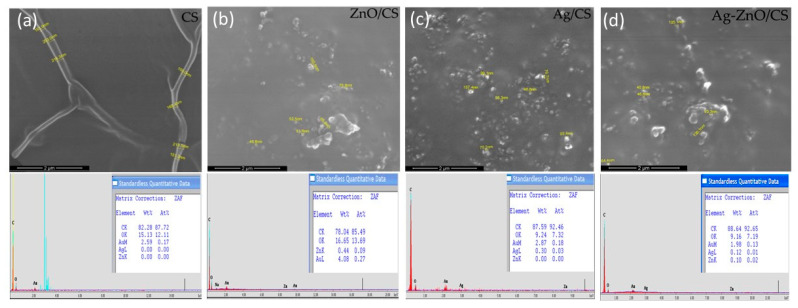
SEM images with EDX of thin films of nanocomposites used for surface modification of SPCE with CS (**a**); ZnO/CS (**b**); Ag/CS (**c**); Ag-ZnO/CS (**d**).

## Data Availability

Data sharing not applicable.

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
