# Peer review of "Cyclic Voltammetry of Screen-Printed Carbon Electrode Coated with Ag-ZnO Nanoparticles in Chitosan Matrix"

_materials, 2023, doi:10.3390/ma16083266_

Round 1

Reviewer 1 Report

The paper "Cyclic voltammetry of screen-printed carbon electrode coated with Ag-ZnO nanoparticles in chitosan matrix" shows comparison between different modification of the surface. Overall, the subject is interested and can be further analyzed to granted extra application. The use of English sometimes is difficult to understand and sentences should be cut for clearer understanding. Beside, some modifications should be updated as follow: 

1. Spacebar should be added before the parentheses and between the value and the unit (example: 1 mg)

2. The citing in the sentences should be unified in the position (after the respected authors or at the end of the sentence).

3. Abbreviations should be used across text as some phrases still in full form despite being reduced beforehand. other abbreviations (for example: HBPT) have no full form.  

4. The diagrams should be redrawn and recalibrated the axes for a better view. Diagram name in the chart is not required. However, the axes should be larger with the full name to be clearer. 

5. Caption of Figure should include caption of a), b), and c) (Figure 1).

6. Figure 5a, b, c, and d should be recollected into one Figure for better comparison. 

7. The synthesis procedure of Ag/CS, ZnO/CS, and Ag-ZnO/CS should be included.

8. The authors should not conclude the agglomeration from optical image but rather SEM images.

9. Since the modification with chitosan is superior in comparison with Ag/CS, ZnO/CS, and Ag-ZnO/CS, explanations should be added regarding the phenomena. 

Author Response

Dear reviewer I attached again the manuscript with modification according to your comments.

Response to Reviewer 1 Comments

Dear reviewer 1

Thank you very much for the interesting comments because helped us in order to clarify and explain better our idea of surface modification for SPCE. As regrading your comments, we have the follows response:

Point 1:

Spacebar should be added before the parentheses and between the value and the unit (example: 1 mg)

Response 1: We checked all the values and units and put the spacebar between the value and unit.

Point 2: The citing in the sentences should be unified in the position (after the respected authors or at the end of the sentence).

Response 2: We changed the citing position at the end of the phrase.

Point 3: Abbreviations should be used across text as some phrases still in full form despite being reduced beforehand. other abbreviations (for example: HBPT) have no full form.  

Response 3: We checked all the abbreviation and for HBPT was a mistake it is HBP-jouse built potentiostat

Point 4: The diagrams should be redrawn and recalibrated the axes for a better view. Diagram name in the chart is not required. However, the axes should be larger with the full name to be clearer. 

Response 4: All the graph has been redrown in origin 7 soft and the name are clear now

Point 5: Caption of Figure should include caption of a), b), and c) (Figure 1).

Response 5: We changed the name of Figure 1 and we included a), b), and c) (Figure 1)

Point 6: Figure 5a, b, c, and d should be recollected into one Figure for better comparison. 

Response 6: We collect all the 3 graph for comparison fig 5a,b,c

Point 7: The synthesis procedure of Ag/CS, ZnO/CS, and Ag-ZnO/CS should be included.

Response 7: The synthesis of is not explained included due to the fact that we included the reference and one of our author did that synthesis. It is possible to be autoplagiat if we present explained the synthesis.

Point 8. The authors should not conclude the agglomeration from optical image but rather SEM images.

Response 8: The optical images are presented because we want to explain the electrical behaviour off the CV response. It is possible that the oxidation and reduction peak are not so clear due to the modified surface of the electrode. Now we are working on improving the viscozity of the deposited solution in order to have a smoothed and better adesive layer on carbon elecgtrode.

Point 9 Since the modification with chitosan is superior in comparison with Ag/CS, ZnO/CS, and Ag-ZnO/CS, explanations should be added regarding the phenomena.

Response 9: We modfied same explination regarding different behavior of electrodes.

Reviewer 2 Report

In this research work entitled “Cyclic voltammetry of screen-printed carbon electrode coated with Ag-ZnO nanoparticles in chitosan matrix”, the authors fabricate and study the nanocomposite chitosan-based systems of zinc oxide (ZnO), silver (Ag) and Ag-ZnO. This is an interesting topic for materials chemistry and sensor technology communities.

This article deals with results that appear not to have been already published. However, I have some comments and concerns about the evaluation of this work, the details are given below.

After reading the paper thoroughly, I could not find any novelty in this work. In the introduction/objective of the study part, the authors stated that the fabricated sensor could apply to detect cancer biomarkers. Technically, there are plenty of cancer biomarkers in our system and present in various biofluids at different concentrations. The authors should focus on this issue and prove their objective.

In the experimental and result in part, I appreciated the home-built potentiostat and modified electrodes work. To provide new knowledge to the existing literature, this work is not sufficient at this stage. Need to explore more electrochemical performance of the fabricated devices as well as an explanation is necessary about the science behind their action.

From this study, I could not find the potential application of the fabricated sensor/electrodes. This study is more or less like a lab report about one feature such as cyclic voltammetry. Must include the study of at least one cancer biomarker detection method either in an artificial sample or a real sample.  There is a lot of space to improve. I am afraid that I can not recommend publishing this manuscript at this stage. 

Author Response

Response to Reviewer 2 Comments

Dear reviewer 2

Thank you very much for the interesting comments because helped us in order to clarify and explain better our idea of surface modification for SPCE. As regrading your comments, we have the follows response:

Point 1:

After reading the paper thoroughly, I could not find any novelty in this work. In the introduction/objective of the study part, the authors stated that the fabricated sensor could apply to detect cancer biomarkers. Technically, there are plenty of cancer biomarkers in our system and present in various biofluids at different concentrations. The authors should focus on this issue and prove their objective.

 Response 1:

The novelty of our work in relation to specialized literature consists of utilization of fresh prepared zinc oxide nanoparticles obtained by hydrolytic method, Ag-ZnO np’s and not commercial one to modify the carbon electrode surface with these systems.Our method is easiest, cheap and the obtaining temperature is under 100 0C. Our systems consists of three antimicrobial agents (Ag NPs, ZnO NPs and Cs) which are chemically bounded to each other.

Point 2:

In the experimental and result in part, I appreciated the home-built potentiostat and modified electrodes work. To provide new knowledge to the existing literature, this work is not sufficient at this stage. Need to explore more electrochemical performance of the fabricated devices as well as an explanation is necessary about the science behind their action.

 Response 2:

Regarding the explore of electrochemical perhormance, this work was and it is a challenge . We are still measuring SPCE modified before to use the cancer cell that we want to measure. The procedure is not so easy an is expensive duet o the price of cell, and we can use the cells only once after defrosting the cells. This measurement are in our plan for present and future measurements.

 Point 3:

From this study, I could not find the potential application of the fabricated sensor/electrodes. This study is more or less like a lab report about one feature such as cyclic voltammetry. Must include the study of at least one cancer biomarker detection method either in an artificial sample or a real sample.  There is a lot of space to improve. I am afraid that I can not recommend publishing this manuscript at this stage. 

Response 3:

Because of the antimicrobial, photolumiscent and photocatalitic activities of developed biocompatible nanocomposites - Ag NPs/Cs, Ag-ZnO NPs/Cs and ZnO/Cs, this materials are promising for applications in the field of cancer cells identification. We are in progress of measurements for cancer cells identification by cyclic voltammetry.

Reviewer 3 Report

The present article entitled “Cyclic voltammetry of screen-printed carbon electrode coated with Ag-ZnO nanoparticles in chitosan matrix " describes the the fabrication of nanocomposite chitosan-based systems of zinc oxide (ZnO), silver (Ag) and Ag-ZnO. Thus, this reviewer recommends the publication of this work in this Journal after addressing the following concerns.

Comments

1.      In abstract section should be provided quantitative information.

2.      The following sentence, the author should provide the suitable references. The composite of graphene oxide/cobalt/chitosan was used in SPCE for electrochemical measurements to determine the various concentration of D-glucose and showed a sensitivity of 15 mM.

3.      2.1 Reagents and Solutions. This section should be provided separately (Materials and reparation method).

4.      The author should perform the following studies for the conformation of metal composites (FTIR, XRD, TEM).

5.       Figure 6. Optical images of surface modified SPCE with CS (a), ZnO/CS (b), Ag/CS (c), Ag-ZnO/CS (d). These images should be provided with higher magnifications.

6.      The conclusion should concise and revised with outstanding point of this work.

7.      Page 8, line 264, what is mean for a good?

8.      Typographical errors and superfluous spaces throughout the manuscript should be corrected.

Author Response

Response to Reviewer 3 Comments

Dear reviewer 3

Thank you very much for the interesting comments because helped us in order to clarify and explain better our idea of surface modification for SPCE. As regrading your comments, we have the follows response:

Point 1:  In abstract section should be provided quantitative information.

Response 1: We checked again the abstract and we added quantitative information regarding the values of anodic and cathodic currents when we have scan rate variation in case of unmodified electrodes.

Point 2:    The following sentence, the author should provide the suitable references. The composite of graphene oxide/cobalt/chitosan was used in SPCE for electrochemical measurements to determine the various concentration of D-glucose and showed a sensitivity of 15 mM.

Response 2: We added the reference.

Point 3

2.1 Reagents and Solutions. This section should be provided separately (Materials and reparation method).

Response 3: Can you please be more explianed with this section?

Point 4:    The author should perform the following studies for the conformation of metal composites (FTIR, XRD, TEM).

Response 4: The nanocomposites used for the modification of electrodes were measured before for another study regardic the antimicrobial activity and int hat article are FTIR and XRD measurements in which is proven the existence of ZnO, Ag, and chitosan, and also the chemical bonds. Maybe TEM analysis will be in our further research regrding the cancer cell identification.

Point 5:     Figure 6. Optical images of surface modified SPCE with CS (a), ZnO/CS (b), Ag/CS (c), Ag-ZnO/CS (d). These images should be provided with higher magnifications.

Response 5: Unfortunately this the maximum maginification for our optical microscope

Point 6:   The conclusion should concise and revised with outstanding point of this work.

Response 6: The conclusions were modified.

Point 7:    Page 8, line 264, what is mean for a good?

Response 7: I changed the phrase.

Point 8:  Typographical errors and superfluous spaces throughout the manuscript should be corrected.

Response 8: We checked again the article for typographical errors and superfluous space.

Round 2

Reviewer 2 Report

Writing should be of better quality. It is necessary to rephrase a few sentences that are difficult to understand. This can do during the copy-editing process. No need to submit for revision again. 

Reviewer 3 Report

The revised work can be accepted in its current form.